# The Identification of Sustainability Assessment Indicators for Road Infrastructure Projects in Tanzania

Chisomo Kapatsa [1], Neema Kavishe [2,*], Godwin Maro [1] and Sam Zulu [2]

1   School of Architecture, Construction Economics and Management (SACEM), Ardhi University,
    Dar es Salaam P.O. Box 35176, Tanzania; godwinmaro2016@gmail.com (G.M.)
2   School of Built Environment, Engineering and Computing, Leeds Beckett University, Leeds LS2 8AG, UK;
    s.zulu@leedsbeckett.ac.uk
*   Correspondence: n.w.kavishe@leedsbeckett.ac.uk

**Abstract:** The performance of sustainability in infrastructure projects continues to face challenges in its implementation and attainment in developing countries, one of which is the lack of appraisal tools and indicators for the assessment of sustainability. Studies indicate that there are no formal indicators for sustainability assessment on road infrastructure projects in Tanzania, the lack of which limits the determination of whether projects implemented are sustainable or not. Therefore, this study aimed at determining the key sustainability assessment indicators used for road infrastructure projects in Tanzania. A concurrent mixed research approach was adopted in which the sample was purposively selected. A content analysis and descriptive statistics using the Statistical Package for the Social Sciences (SPSS 20.0) were used to analyze qualitative and quantitative data, respectively. The findings indicate that 24 indicators are applicable to Tanzania. Among the highly ranked include "health and safety training to workers", "health and safety personnel in the project team", "site barriers and safety warning signs", "personal protective equipment (PPE) provision", and "waste collection". The qualitative results further support the identified sustainability assessment indicators on road infrastructure projects in Tanzania, with one new indicator of "air quality" emerging. The findings inform the government and other relevant stakeholders in the construction industry including planners, designers, and project managers of the key sustainability assessment indicators for roads, which would influence regulation as well as policies to improve the sustainability performance of road projects in Tanzania.

**Keywords:** sustainability; road infrastructure projects; assessment; sustainability indicators; Tanzania

## 1. Introduction

Sustainability phenomena began decades ago, intending to balance the three sustainability aspects, environmental, economic, and social, in project delivery and to minimize the impacts on the environment [1]. Over the years, the sustainability concept has been established as a notion to be abided by, for sustainable development [2]. The construction sector has a significant role in sustainable development through the provision of physical infrastructure to meet the needs of humanity. However, it also has negative effects on the environment and society [3]. The traditional method of infrastructure project delivery is not sustainable due to its heavy reliance on natural resources, emissions, noise pollution waste generation, habitat loss, and fragmentation [4,5], which translates to poor sustainable development, despite its positive impacts on the economy and regional developments [6]. The aforesaid situation has increased the demand and emphasis on sustainability in infrastructure project delivery to enhance sustainable development [7].

Sustainable development entails meeting current generations' needs and aspirations while not jeopardizing required resources by subsequent generations to fulfill their requirements [8]. It is an integration of three dimensions widely known as the Triple Bottom Line (TBL) which includes environmental, social, and economic dimensions [9]. However, other dimensions, in addition to these three, are recognized, for instance, the technological dimension [10] and institutional dimension [11]. Nevertheless, the three dimensions are prominent [3,12–14].

The sustainability performance of infrastructure projects is essential to achieving sustainable development because it allows for the assessment of how well the projects are being carried out, to gauge their sustainability performance, and, if necessary, to identify areas for improvement [15]. As a result, sustainability should be monitored and assessed [16]. For this, sustainability assessment appraisal tools and techniques as well as indicators are used [16]. Nevertheless, there is a lack of sustainability appraisal tools and indicators [17]. The lack of which limits the determination of whether projects implemented are sustainable or not. The authors of [18] showed that most construction businesses find it challenging and subjective to determine their sustainability level's direction; nevertheless, employing sustainability indicators can be conducted as a potential remedy for this. According to [19], the development of indicators will promote and enhance sustainability evaluations. The authors of both [11,20] acknowledge the need to identify indicators for sustainability assessments in all project phases and from the various sustainability dimensions. Correspondingly, Refs. [10,21] established indicators based on the lifecycle of projects, Ref. [22] on the planning phase, and [23] on the execution phase. Nonetheless, the indicators cannot fully be universally adaptable due to various constraints, needs, national objectives, policies, and priorities [21]; therefore, context-specific indicators are required. Despite numerous studies on sustainability in Tanzania, little focus has been placed on how sustainability is assessed on projects including road infrastructure projects. As a result, this study sought to identify the environmental and social sustainability assessment indicators used for road infrastructure projects.

## 2. Literature Review

According to [24], a sustainability indicator is a quantifiable or qualitative characteristic that indicates a desired result directed toward achieving sustainability goals. The authors of [1] describe a sustainability indicator as a measurable component of social, economic, or environmental systems that makes it possible to track and monitor changes on a project. Moreover, a good indicator should be relevant, reliable, based on accessible data, and easy to understand [25]. Moreover, sustainability indicators should be identified for all dimensions [1]. Nonetheless, Ref. [26] opines that the environmental aspect is favored more than the rest of the aspects. As an emphasis, Ref. [27] opines that the social aspect is ignored in most construction projects and emphasizes the inclusion of the social aspect in sustainable construction projects.

Furthermore, sustainability indicators should apply to all construction project phases, i.e., the inception, planning, execution, operating, and demolition stages [3]. Various studies have identified these indicators in accordance with the project phases; however, the design and execution phases are the focus of this study because attention to these processes will result in a more sustainable product. Various indicators have been identified in different places; Table 1 illustrates a summary of the previous studies on sustainability assessment indicators.

**Table 1.** Summary of previous research studies on sustainability assessment indicators.

| Reference | Country | Findings |
|---|---|---|
| [10] | Australia | The framework comprises a reduced set of 16 indicators. The key indicators were enhancing physical health, adjustments to the climate, mental health, water-related regulation, and air quality. |
| [11] | Sweden | Efficiency, safety, accessibility, livability, emissions, and resource use were the main indicators identified in the study. |
| [12] | Spain | Framework included 30 indicators based on the three dimensions with the following as the highly ranked: usage of energy, handling of waste, environmental footprint, emissions of carbon dioxide ($CO_2$), and health and safety. |
| [15] | Taiwan | A total of 31 indicators were used in the framework; key indicators include materials, water efficiency, biodiversity protection, health and safety, and land use. |
| [17] | South Africa | Based on six dimensions of environmental, social, economical, resource utilization, health and safety, and project management, 30 indicators emerged. Indicators under health and safety were highly ranked; however, the following indicators' rankings were significant; project duration, lifecycle cost, environmental assessment impact, availability of materials, and a health and safety management system. |
| [19] | China | The top emerging indicators were an examination of market supply and demand, financial risk, public safety, the impact of local developments, water quality, and the impact of land pollution. |
| [21] | Taiwan | Among the three dimensions, the environmental aspect was highly ranked, with key indicators including, environmental protection, pollution reduction, and resource usage, which are all viewed as crucial under the environmental aspect. From the social aspect, the quality of life of mankind is significant, and from the economic aspect, eco-economics is significant. |
| [22] | South Africa | The rising common indicators were user acceptability, financial management aspects, safety and security, and infrastructure conditions and impacts. |
| [25] | Iran | The top indicators emerged as follows: air pollution, safety, fuel consumption, and green space destruction. |
| [28] | Nigeria | Indicators were based on the three dimensions. The top environmental indicators were biodiversity and energy use. The social dimension's educational component, stakeholder fairness, and health and wellbeing are all included. The economic dimension comprises a low cost of maintenance, long-term costing, and local economic growth. |
| [29] | Greece | Indicators from the environmental dimension were highly ranked. The key indicators include sustainable materials, ecological efficiency, efficient energy consumption, environmental impact assessment reports, and efficiency in resource allocation. |

Table 1 illustrates that most indicators are divided according to dimensions. In addition, there are some similarities in the indicators identified; however, principally, the indicators identified are different in the various contexts. Correspondingly, not all indicators can apply to a specific context, and inherently, a selection criterion for indicators that apply to a particular study is ideal [11]. The criteria include relevance to the local context [30], the understandability of indicators [5], the availability of data [2], and a limited number of indicators [11]; however, no crucial indicators may be omitted. Correspondingly, Ref. [31] suggests that 20 to 30 are manageable for a study.

In addition, Ref. [32] states that it is helpful to categorize sustainability indicators according to clearly defined categories and sub-categories to enable the selection of indicators for certain applications. The authors of [21] established the sustainability dimensions as the main categories of the indicators, whereas the authors of [15] established their own categories centered on the emerging sustainability issues under the sustainability dimensions.

Similarly, in this study, categories were identified based on the trend and emerging sustainability issues. As a result, as in [21], the sustainability dimensions were considered as the main categories in this study, from which three sub-categories based on emerging sustainability issues emerged, namely, resource utilization, health and safety, and social equity and justice. A further breakdown of seven specific indicators, including energy efficiency, water efficiency, sustainable materials, waste management, health and safety,

employment, and stakeholder involvement, from which 27 indicators were identified, was established, as depicted in Table 2. This study intended to find significant indicators in Tanzania from the 27 indicators.

**Table 2.** Sustainability assessment indicators.

| Category | Sub-Category | Indicators | Evaluation Items |
|---|---|---|---|
| Environmental | Resource utilization | Energy efficiency | • Use of renewable energy like solar and wind.<br>• Energy-saving initiatives, e.g., energy-saver bulbs for lighting.<br>• Use of energy-efficient (diesel or electric powered) plants and machinery and energy-efficient appliances such as air conditioners.<br>• Energy consumption monitoring plan. |
| | | Water efficiency | • Grey water reuse.<br>• Control and monitoring plan for water.<br>• Storm water runoff management.<br>• Rainwater harvesting. |
| | | Sustainable materials and resources | • Recyclable materials.<br>• Use of natural resources like soil, wood, bamboo, straw, etc.<br>• Durable material resources.<br>• Locally available materials. |
| | | Waste production and management | • Waste collection.<br>• Recycling and reuse of waste.<br>• Use of waste-reducing construction techniques (for example, precast, modularization). |
| Social | Safety | Health and safety | • Health and safety training for workers (e.g., tool box talks).<br>• Inclusion of health and safety personnel in the project team.<br>• Provision of personal protective equipment (PPE).<br>• Onsite medical services like first aid kits.<br>• Site hoarding and safety warning signs. |
| | Social equity and justice | Employment | • Local labor.<br>• Employee well-being and benefits.<br>• Good working conditions.<br>• Gender equality. |
| | | Stakeholder involvement | • Acceptability of stakeholders.<br>• Early involvement of contractors and suppliers.<br>• Training and knowledge transfer in regard to the project. |

## 3. Methodology

### 3.1. Research Design

The concurrent mixed approach was adopted in this study to achieve the main objective. In relation to [33], this approach allows for the offset of the strengths and weaknesses of either method. Additionally, triangulation and complementarity benefits suffice, which enhance the reliability of the results [34]. Moreover, concurrent mixed approaches, as emphasized by [35], combine qualitative and quantitative data-gathering methods for a simultaneous data analysis. Accordingly, in this study, in parallel and with equal standing, both qualitative (semi structured interviews) and quantitative (a questionnaire survey) approaches were used. Consequently, this contemporaneous approach made it possible to use the findings of one method to support the conclusions of the other regarding a particular phenomenon [36].

A multi-case study research design was adopted in order to gain a deeper understanding of the subject matter [37]. This is also in line with [38] which notes that case studies provide much deeper results as opposed to other methods. Similarly, Ref. [39] argues that

the case study research approach is suitable for providing a detailed understanding of issues with little research. Consequently, consistent with [37], the case study approach was chosen for the purpose of gaining deeper insights and for the understanding of the key sustainability assessment indicators for road infrastructure projects implemented in Tanzania because little research has been undertaken in this topic. Furthermore, it was important to use data from multiple sources to present more holistic and realistic information since this study involved multiple stakeholders with different experiences and perspectives.

*3.2. Case Selection*

This study selected 3 road infrastructure projects within the Dar es Salaam region in Tanzania. The research was conducted in Dar es Salaam, Tanzania, because it is a fast-growing city, having a diverse social, cultural, and economic environment. Also, it has many ongoing road infrastructure projects, as evidenced by the Tanzanian Road Authority (TANROADS) website. To select the projects with relevant and reliable data, the following selection criteria were used:

1.  Nature of the projects: only road infrastructure projects were selected because this study mainly focused on road infrastructure projects instead of building projects, because, in consideration of the distance they cover, during project execution, they have massive social and environmental effects.
2.  Size of the projects: only large construction projects (ranging from TSh 50 billion, equivalent to GBP 15.7 million, and above) were selected, because it is usually the large projects that consider sustainability as compared to small projects.
3.  Ongoing or recently completed projects: both ongoing and completed projects within 3 years of completion (2021–2023) were selected, as they helped to obtain recent data on the practice, which can portray a real picture of the current practice on the ground.
4.  Funding source: Only donor-funded projects were selected because they usually have sustainability requirements. Sustainability requirements were crucial in the selection process of the case studies in order to obtain the relevant information for this study.

The list of road projects was sourced from the TANROADS website, which yielded 182 road infrastructure projects. The search was narrowed down to road projects based in Dar es Salaam, both completed and ongoing, from which 28 projects emerged. After applying the case study selection criteria, 2 projects were selected. Another source was the JICA website, from which a total of 7 projects were identified. Then, based on the above detailed 4 selection criteria, one project qualified; therefore, the total number of selected projects was 3, as depicted in Table 3 below.

**Table 3.** Selected case studies.

| Cases | Project Name | Distance | Estimated Cost (Tsh) | Client | Project Status |
|-------|-------------|----------|---------------------|--------|----------------|
| Case 1 | Bus Rapid Transport (BRT) phase 2 lot 1 | 20 km | 198.4 billion | TANROADS | Ongoing |
| Case 2 | Construction of Kijazi Interchange | 5.95 km | 177.2 billion | TANROADS | Completed (2022) |
| Case 3 | Widening of new Bagamoyo road phase 2 | 4.3 km | 71.8 billion | TANROADS | Completed (2021) |

*3.3. Population*

From the selected cases, data were collected to identify the main sustainability assessment indicators for road infrastructure projects in Tanzania, as identified from the literature. The population for this study is in two categories. The first group includes project participants from the selected case studies, mainly the key players. The second group includes experts in sustainability as academicians, experts from regulatory authorities, and the client and funding agencies. The purposive sampling technique was used to select the key players in the selected projects including (i) project managers, (ii) architects, (iii) engineers, (iv) quantity surveyors, (v) environmental specialists, and (vi) social

workers that are directly involved in the project both from the consultant and contractor's side, and the financer representative of each project was sampled. In addition, the client representative from TANROADS, as well as experts from the regulatory authorities, National Environment Management Council (NEMC) and Tanzania Green Building Council (TGBC), and Tanzanian researchers on sustainability were involved. In total, 47 respondents were identified.

### 3.4. Questionnaire Survey Administration

The university administration provided ethical clearance prior to data collection. The data were collected through questionnaires and semi structured interviews. The mixed-method approach was preferred because it maximizes the benefits of both approaches while minimizing their drawbacks [33]. The questionnaires were distributed by hand as well as online using Google Forms between January 2023 and February 2023. The questionnaire comprised close-ended questions and was in 2 parts. Part A comprised demographic information whereas part B required respondents to rate the significance of the 27 indicators established in the literature, which were used for assessing sustainability on road infrastructure projects using a 5-point Likert scale, where 1 = strongly disagrees, 2 = disagrees, 3 = neutral, 4 = agrees, and 5 = strongly agrees. Out of the 47 questionnaires dispersed, only 30 questionnaires were returned; nevertheless, one (1) was not completed, hence only 29 were deemed legitimate, representing a 61% response rate. A total of 29 questionnaire survey participants may seem like an insignificant sample size. However, given that it comprised more than 50% of the sample population, this response rate was sufficient [10].

### 3.5. Interviews

Semi structured interviews were conducted with the respondents from the case studies selected, specifically top-ranking officials, including personnel from contractors, consultants, clients (TANROADS), funding agencies, NEMC, TGBC, and academicians. The interviews were conducted in Dar es Salaam, Tanzania, between January and February 2023.

In accordance with [40], semi structured interviews were opted for due to their ability to produce precise information and their flexibility in helping to explore new perspectives on issues that are not predetermined in the study. The respondents were purposively selected, and willingness to participate and easy reach were considered as well. In total, 11 interviews were conducted. The interview took approximately 20 to 30 min. According to [34], which recommends 30 to 60 min as an acceptable time, the amount of time spent in the interview appears reasonable. The structure of the interviews was similar to the structure of the questionnaire.

### 3.6. Data Analysis

Using the SPSS package and Microsoft Excel software, the quantitative data acquired for this study were analyzed using descriptive statistics from which measures of central tendency, specifically mean values, standard deviation, and Relative Importance Index (RII), were produced. The RII scores were used to rank the indicators in ascending order. Meanwhile, the qualitative data were analyzed using the content analysis technique, specifically the summative approach, which focuses on identifying key words and subject frequencies and recurrences [41]. Moreover, this is a good approach when trying to find out the opinions, knowledge, and views of people from a set of variables, which is the case in this study. The data collected were coded, in the sense that the text or words from the interviewees were scrutinized to establish a single or a few words that represent the main point from the text. Then, frequencies were assigned based on the number of respondents to one point.

## 4. Findings and Discussion

### 4.1. Questionnaire Respondent's Profile

Table 4 summarizes the personal background information that was obtained. With respect to experience, the results indicate that more than half, 16 out of 29 (55%), had more than 5 years of work experience. The data were collected from different organizational positions in the construction industry, which allows for the inclusion of different organizations' perspectives on sustainability aspects. In terms of profession, the participants were of different professional backgrounds; however, 18 out of 29 (62%) were engineers, which is not surprising because most road projects are usually undertaken by Civil Engineers. The participants were from different organizations, with the majority, 14 of 29 (48%), being from contracting firms. For educational qualifications, the majority, 27 out of 29, had a minimum of a bachelor's degree, and only 2 had a Diploma. As noted in [28], the respondents were determined to have the necessary experience, qualifications, and expertise to offer accurate and reliable data for this study based on their demographic attributes.

**Table 4.** Demographic information of the participants.

| Characteristics | Frequency | Percent |
|---|---|---|
| **Experience** | | |
| ≤5 years | 13 | 44.8 |
| 6–10 years | 9 | 31.0 |
| 11–15 years | 4 | 13.8 |
| >15 years | 3 | 10.3 |
| Total | 29 | 100.0 |
| **Organization Position in the construction industry** | | |
| Client | 4 | 13.8 |
| Consultant | 10 | 34.5 |
| Contractor | 14 | 48.3 |
| Financer | 1 | 3.4 |
| Total | 29 | 100.0 |
| **Profession of respondent** | | |
| Project Manager | 2 | 6.9 |
| Engineer | 18 | 62.1 |
| Quantity Surveyor | 1 | 3.4 |
| Environmental Specialist | 2 | 6.9 |
| Community Social worker | 2 | 6.9 |
| Any other | 4 | 13.8 |
| Total | 29 | 100.0 |
| **Education level of respondent** | | |
| Diploma | 2 | 6.9 |
| Bachelor's degree | 22 | 75.9 |
| Master's degree | 4 | 13.8 |
| PHD | 1 | 3.4 |
| Total | 29 | 100.0 |

**Notes:** The following is a breakdown of the 'any other' professions: Administrative Managers (1), Health and Safety officer (2), and Translator (1). Since they were not among the choices provided in the survey questionnaire, the respondents provided these professional roles separately.

### 4.2. Interviewees' Profile

Table 5 displays the interviewees' characteristics, including experience, organization position, title within their organizations, education level, and the participant's professional background. Table 5 illustrates that the majority of the interviewees, (n = 8) 73%, fell within 6–10 years, 11–15 years, and >15 years, which implies that the majority of the interviewees have more than 5 years' experience. Moreover, based on the respondent's reported demographic background, it can be seen that all the major players in sustainable infrastructure projects, from various management levels, participated in the interviews, which increases the validity and reliability of the results. According to [42], the sample size

required for interviews to achieve the saturation point is between 5 and 50. Therefore, the sample size (n = 11) and level of responsiveness are regarded as sufficient.

**Table 5.** Interviewee profile.

| Interviewees | Experience | Role | Position | Educational Level | Profession |
|---|---|---|---|---|---|
| A | 6–10 years | Regulatory authority | Project manager | Bachelor's degree | Architect |
| B | 6–10 years | Consultant | Environmental Engineer | Bachelor's degree | Environmental Specialist |
| C | ≤5 years | Client | Planning Engineer | Bachelor's degree | Civil Engineer |
| D | ≤5 years | Financer | Admin. manager | Master's degree | Administrator |
| E | 11–15 years | Contractor | Project Engineer | Bachelor's degree | Civil Engineer |
| F | 11–15 years | Regulatory authority | Regional Manager | Master's degree | Environmental Specialist |
| G | >15 years | Financer | Senior Transport Specialist | Master's degree | Civil Engineer |
| H | ≤5 years | Regulatory authority | Environmental Specialist | Bachelor's degree | Environmental Specialist |
| I | 11–15 years | Contractor | Project Quantity Surveyor | Bachelor's degree | Quantity Surveyor |
| J | 11–15 years | Consultant | Project Engineer | Bachelor's degree | Civil Engineer |
| K | >15 years | Academician | Senior Lecturer | PHD | Quantity Surveyor |

*4.3. Ranking of Sustainability Assessment Indicators*

Table 6 shows the summary of the results obtained from descriptive statistics on the 27 sustainability assessment indicators as well as their rankings. The results of the descriptive statistics, such as mean, standard deviation, and RII, are illustrated. The indicators were ranked in ascending order based on the RII scores which range between 0 and 1, with greater values signifying higher rankings and lower scores signifying lower rankings. As cited by [43], the RII resulting values are classified as low (L) ($0 \leq \text{RII} \leq 0.2$), medium-low (M–L) ($0.2 \leq \text{RII} \leq 0.4$), medium (M) ($0.4 \leq \text{RII} \leq 0.6$), high-medium (H-M) ($0.6 \leq \text{RII} \leq 0.8$), and high (H) ($0.8 \leq \text{RII} \leq 1$). Additionally, based on the mean scores, the indicator's significance was determined. Since a 5-point Likert scale was employed, 1 = strongly disagree and 5 = strongly agree, with a mid-value of 3 = neutral, a mean score of "3.5" or more than average would indicate that a statement was more frequently applicable, and a score below "3.5" would indicate that it was less applicable, as noted in [17].

As illustrated in Table 6, 23 out of 27 indicators were considered significant in the assessment of sustainability on road infrastructure projects. In addition, the mean values ranged from 4.59 to 2.97. Since the mean values of the top 23 indicators were above the cut-off point of a mean value of 3.5, this shows that they are important in the assessment of sustainability on road infrastructure projects. In the following sections, the indicators are discussed.

4.3.1. Energy Efficiency

The energy efficiency group consists of four sustainable indicators. Generally, the findings show that the indicators in this group are lowly ranked. The use of energy efficient plants and machinery during construction is highly ranked in this category with a mean score of 3.83 and an RII of 0.766, followed by the use of renewable energy with a mean score of 3.62 and an RII of 0.724. Energy-saving initiatives was the least ranked with a mean of 3.48 which is below the cut-off point of 3.5. These results are similar to [7,19], where energy saving and the use of renewable energy are lowly ranked. This implies that energy efficiency is not highly considered in road infrastructure projects in Tanzania, which is contrary to [12,28], where energy-saving initiatives were highly prioritized in assessing sustainability. As a result, there is a need for more emphasis and consideration on energy-saving initiatives in road infrastructure projects.

**Table 6.** Summary of sustainability assessment indicators.

| Sustainability Indicators | Mean | RII | Standard Deviation | Rank | Significance |
|---|---|---|---|---|---|
| **ENVIRONMENTAL** | | | | | |
| **Energy Efficiency** | | | | | |
| Use of renewable energy like solar and wind | 3.62 | 0.724 | 1.374 | 20 | Yes |
| Energy-saving initiatives | 3.48 | 0.696 | 1.214 | 24 | No |
| Use of energy-efficient plants, machinery, and appliances | 3.83 | 0.766 | 1.167 | 12 | Yes |
| Energy-consumption monitoring plan | 3.55 | 0.710 | 1.088 | 21 | Yes |
| **Water Efficiency** | | | | | |
| Grey water reuse | 2.97 | 0.594 | 1.017 | 27 | No |
| Control and monitoring plan for water | 3.52 | 0.704 | 0.986 | 22 | Yes |
| Storm water run-off management | 3.79 | 0.758 | 1.177 | 14 | Yes |
| Rainwater harvesting | 3.24 | 0.648 | 1.154 | 26 | No |
| **Sustainable Materials and Resources** | | | | | |
| Recyclable materials | 3.38 | 0.676 | 1.015 | 25 | No |
| Use of natural resources | 3.76 | 0.752 | 0.872 | 15 | Yes |
| High-quality durable materials | 4.00 | 0.800 | 0.964 | 9 | Yes |
| Locally available materials | 3.79 | 0.758 | 0.940 | 13 | Yes |
| **Waste Production and Management** | | | | | |
| Waste collection | 4.24 | 0.848 | 1.023 | 5 | Yes |
| Recycling and reuse of waste | 3.62 | 0.724 | 1.178 | 19 | Yes |
| Use of waste-reducing construction techniques | 3.52 | 0.704 | 1.122 | 23 | Yes |
| **SOCIAL** | | | | | |
| **Health and Safety** | | | | | |
| Health and safety training for workers | 4.59 | 0.918 | 0.867 | 1 | Yes |
| Inclusion of health and safety personnel in the project team | 4.48 | 0.896 | 0.871 | 2 | Yes |
| Provision of PPE | 4.31 | 0.862 | 0.891 | 4 | Yes |
| Onsite medical services | 4.21 | 0.842 | 0.940 | 6 | Yes |
| Site hoarding and safety warning signs | 4.38 | 0.876 | 0.979 | 3 | Yes |
| **Employment** | | | | | |
| Local labor | 4.14 | 0.828 | 0.990 | 7 | Yes |
| Employee well-being and benefits | 3.97 | 0.794 | 1.052 | 10 | Yes |
| Good working conditions | 3.83 | 0.766 | 1.002 | 11 | Yes |
| Gender equality | 3.69 | 0.738 | 1.039 | 17 | Yes |
| **Stakeholder Involvement** | | | | | |
| Acceptability and engagement of stakeholders | 3.76 | 0.752 | 1.023 | 16 | Yes |
| Early involvement of contractors and suppliers | 3.62 | 0.724 | 1.015 | 18 | Yes |
| Training and knowledge transfer to stakeholders | 4.14 | 0.828 | 0.990 | 8 | Yes |

**Notes**: Mean score based on a 5-point Likert scale. RII is the Relative Importance Index; Ranking is based on the RII, where the higher the RII value the more important the variable. In the case of having similar RII values, the indicators with a lower standard deviation are ranked higher.

4.3.2. Water Efficiency

Four indicators make up the water efficiency group, and the results show that this category is generally poorly ranked. Nonetheless, storm water runoff management has the

highest ranking with a mean score of 3.79 and an RII of 0.758, followed by water control and monitoring plan with a mean score of 3.52. On the other hand, rainwater harvesting and grey water reuse were least ranked, both with mean scores below the cut-off point. This suggests that, despite their importance in attaining sustainability, water-saving methods like rainwater harvesting and grey water reuse are rarely given much thought in road infrastructure projects. The results, however, are contrary to [28] which asserts that water conservation measures must be put in place to reduce water consumption in infrastructure projects because they also help to reduce overall project costs and support biodiversity protection, in accordance with the 2007 European Union Sustainability Policy. A greater effort is needed to ensure water efficiency when implementing road infrastructure projects.

### 4.3.3. Sustainable Materials and Resources

There are four indicators in the group of sustainable resources and materials. The findings indicate that the use of high-quality, durable materials is highly ranked with a mean score of 4.00 and an RII of 0.800. The use of local materials and use of natural resources have mean scores of 3.79 and 3.76 and an RII of 0.758 and 0.752, respectively. In support of these results, Ref. [44] contends that the usage of long-lasting, quality, and durable materials that require less maintenance in the long run and a top-notch operating and maintenance system are helpful in the attainment of sustainability. Moreover, Ref. [45] asserts that using natural and locally accessible materials is the best form of resource usage because it helps save materials. Unexpectedly, the use of recyclable materials is lowly ranked below the cut-off limit, with a mean value of 3.38 and an RII of 0.678, contrary to [4]'s assertion that the use of recyclable materials in a project plays a significant role in sustainability. Despite this, the results typically suggest that resource usage and preservation are considered and put into practice in road infrastructure projects.

### 4.3.4. Waste Production and Management

The waste production and management group has three indicators. According to the results, waste collection is ranked the highest with a mean score of 4.24 and an RII of 0.848. The mean score is relatively higher than the cut-off point of 3.5 which shows that waste collection is highly considered in road infrastructure projects. This finding is similar to [12,46] which state that the collection and disposal of waste-like debris, leftover or unwanted materials, is a crucial aspect in waste reduction on sites. The recycling and reuse of waste as well as the use of waste-reducing techniques were also ranked highly above the cut-off point with mean scores of 3.62 and 3.52, respectively, which shows that they are moderately considered in road infrastructure projects, comparable to [47] which indicates that techniques like modularization and the use of precast concrete are ideal for waste reduction. The results imply that waste management techniques are highly regarded.

### 4.3.5. Health and Safety

Five indicators make up the health and safety category. These indicators were highly ranked overall, with mean values ranging from 4.21 to 4.59. However, health and safety training for workers was the highest ranked indicator out of all. These results imply that health and safety issues are of paramount importance and are highly implemented in road infrastructure projects and that there are high levels of awareness of health and safety matters in the construction industry. These results can be attributed to the enforcement and regulation of health and safety issues by OSHA, as well as the strict demands from funders, as this is one of the requirements in project implementation. These findings are comparable to those of [10,48] which place a significant priority on health and safety issues. This is similar to [49] which asserts that efforts like the provision of PPE, health and safety trainings, site hoardings, warning signs, and the presence of health and safety personnel on site are essential to guarantee the safety of the community and the workers.

### 4.3.6. Employment

The employment group consists of four indicators. The employment of local labour is ranked highly with a mean score of 4.14 and an RII of 0.828. This is similar to [45] which contends that infrastructure projects should create employment for local inhabitants, resulting in the betterment of their lives and permitting internal money circulation across the nation, consequently raising the quality of life and promoting the local economy and promoting the development of sustainable infrastructure [50,51]. Other indicators, such as 'employee well-being and benefits', 'good working conditions', and 'gender equality', were also considered highly. In favor of this, Ref. [52] argues that, in addition to providing employment to local residents, employees must be encouraged or inspired through fairness and decent workplace practices, such as appropriate working hours and adequate breaks. The inclusion of women in project teams is another crucial factor, which is often overlooked yet is crucial for improving sustainable infrastructure [53]. Nonetheless, the results indicate that employment aspects are given consideration in road infrastructure projects, hence enhancing sustainability.

### 4.3.7. Stakeholder Involvement and Engagement

Stakeholder involvement and engagement as a category has three indicators. Given that all the indicators in this group had mean values between 3.62 and 4.14, which were all above the cut-off limit, they were all considered significant. Training and knowledge transfer to stakeholders was ranked highly in this group. These findings are similar to [15,52]. This indicator relates to increasing an organization's capacity to improve personnel competencies in order to improve project performance. Additionally, trainings for third parties, in particular the community, are crucial for raising the awareness of projects, the effects of construction activities, and infrastructure in the community [54]. The acceptability and engagement of stakeholders and the early involvement of contractors and suppliers as indicators are also considered. According to [45], effective stakeholder management techniques are essential to ensure stakeholders' acceptability and to address any potential issues, so early involvement of all project stakeholders in decision making as well as information sharing is crucial. Overall, these findings suggest that stakeholders are held in high esteem in road infrastructure projects, which is laudables because it is one of the most important ways to increase social sustainability.

### 4.4. Interview Results

To improve the validity of the results, the quantitative and qualitative results were triangulated. A question on the sustainability indicators considered in road infrastructure projects was posed. The responses were coded, from which 27 indicators emerged. Similar to [17], indicators with a frequency above the midpoint value (5.5) were considered significant for this study. As depicted in Table 7, seven indicators were considered applicable.

The responses from the interviews indicate that waste production and management is highly considered in sustainable infrastructure projects, including reduction in waste production as well as waste collection and disposal. For instance, interviewee **I** stated that "*a waste management plan goes along way, planning on waste segregation, how often waste is collected as well as identification of proper places for disposal is a crucial element of sustainable infrastructure projects*".

These results correspond to the quantitative results where this indicator was ranked the highest. The provision of PPE, provision of barriers and safety signs, and health and safety training for workers were highly ranked as well, which is in line with the results from the questionnaire survey. This clearly indicates the importance of the implementation of health and safety measures in infrastructure projects. Most of the interviewees reported that the adherence to health and safety in projects helps enhance project performance in preventing delays that may arise from lost time in addition to non-productivity arising from accidents. Other indicators considered include storm water runoff management and stakeholder involvement, in line with the quantitative data as well. However, a new

indicator of air quality emerged from the interviews. The responses from the interviewees indicate that the control of dust as well as hazardous odor from the construction sites have to be considered to enhance air quality in areas surrounding the projects.

**Table 7.** Sustainability assessment indicators (interviewee perspective).

| No. | Sustainability Indicators | Interviewees | | | | | | | | | | | Frequency |
|---|---|---|---|---|---|---|---|---|---|---|---|---|---|
| | | A | B | C | D | E | F | G | H | I | J | K | |
| | **Environmental** | | | | | | | | | | | | |
| 1 | Waste management (collection and disposal) | √ | √ | √ | √ | √ | √ | | | √ | √ | √ | **9** |
| 2 | Local materials | √ | | √ | | | | | | | √ | | 3 |
| 3 | Grey water reuse | √ | | | | | | | | | | | 1 |
| 4 | Energy Saving | √ | | | | | | √ | | | | √ | 3 |
| 5 | Energy-efficient machines and plants | | √ | | | | | √ | | | | | 2 |
| 6 | Air quality | | √ | √ | √ | | √ | √ | | | | √ | **6** |
| 7 | Water pollution | | √ | | | | √ | | | | | | 2 |
| 8 | Sustainable materials (recyclable and natural materials) | | √ | √ | | | √ | | | | √ | √ | 5 |
| 9 | Storm water runoff | | √ | √ | √ | √ | | | √ | | √ | | **6** |
| 10 | Noise pollution | | | | √ | | √ | | | | √ | √ | 4 |
| 11 | Efficiency water usage | | | | | | | | | | | √ | 1 |
| | **Social** | | | | | | | | | | | | |
| 12 | Durability | √ | | | | | | | | | | | 1 |
| 13 | Non-motorized transport (pedestrians and Cycling) | | √ | | | | | | | | | | 1 |
| 14 | Training on health and safety | | √ | √ | √ | √ | √ | √ | | √ | | | **7** |
| 15 | Provision of PPE | | √ | √ | √ | | √ | √ | √ | √ | | √ | **8** |
| 16 | Good condition of machines | | √ | | | | | | | | | | 1 |
| 17 | Employee well being | | √ | | √ | | √ | √ | | | | | 4 |
| 18 | Training and knowledge transfer | | √ | | | √ | | | | √ | √ | | 4 |
| 19 | Stakeholder involvement | | √ | | √ | √ | | | √ | √ | | √ | **6** |
| 20 | Gender equality | | | √ | √ | | | | | | | | 2 |
| 21 | Barriers and safety signs | | √ | | | | √ | √ | √ | √ | √ | √ | **8** |
| 22 | Local labor | | | | √ | | √ | | | √ | √ | | 4 |
| 23 | Compensation for displacement | | | | √ | | | √ | | | | √ | 3 |
| 24 | Provision of temporary roads | | | | | √ | | | | | √ | √ | 3 |
| 25 | Corporate Social responsibility | | | | | √ | | | | √ | | √ | 3 |
| 26 | Reduction of traffic congestion | | | | | | | √ | | | | | 1 |
| 27 | Good communication among stakeholders | | | | | | | √ | | | | | 1 |

**Notes**: The applicable indicators are bolded.

## 5. Conclusions

Considering the negative repercussions arising from infrastructure projects on the environment and society, there is a need for sustainability in the infrastructure project delivery. Using assessment indicators to gauge sustainability's performance is one strategy to improve it. Even though sustainability indicators are identified elsewhere, context-specific indicators are essential. Consequently, this study investigated the sustainability assessment indicators on road infrastructure projects in Tanzania. With 27 indicators determined from the literature that are derived from the environmental and social dimensions, 23 indicators from the quantitative results are applicable to Tanzania. Seven groupings were created to categorize these indicators. From the seven categories, the top indicators are from the health and safety group, which are health and safety training for workers, the inclusion of health and safety personnel in the project team, site barriers and safety warnings, and the provision of PPE. Other top indicators include waste collection, onsite medical services, the use of local labor, training and knowledge transfer to stakeholders, and the use of high-quality, durable materials. The water efficiency category, however, has the least-weighted indicators, namely, grey water reuse and rainwater harvesting.

Likewise, the findings from the interviews identified seven indicators, most of which were similar to those identified from the questionnaire survey, which emerged as applicable to Tanzania, validating the results collected from the quantitative survey. Nonetheless, from the interviews, a new indicator of air quality emerged, and thus the adoption of the mixed method provided an opportunity to explore some of the unidentified indicators. So

far, the findings indicate that there is a good understanding and awareness of sustainability measures, and they are generally incorporated and implemented in road infrastructure projects in Tanzania. Nevertheless, it was noted that the least measures of protecting and maintaining water sources during road infrastructure project developments are considered, and as a result, more efforts in improving and implementing water-saving initiatives in road infrastructure projects need to be put in place.

This study has developed indicators to be used for sustainability assessments on road infrastructure projects in Tanzania during the planning and execution phases in order to guarantee that the projects are carried out in a sustainable manner. The government and all other relevant stakeholders in the construction industry, including planners, designers, project managers, and environment experts, are informed of the key sustainability assessment indicators for roads, which would influence regulations as well as policies to improve the sustainability performance of their road projects as well as inform the implementation and assessment of the sustainability of road projects in order to facilitate achieving the SDGs 2030.

**Author Contributions:** C.K. initiated and conceptualized the research project, carried out the mixed-methods empirical research, contributed to the literature review, and drafted the manuscript. N.K., G.M. and S.Z. supervised the research project and contributed to the methodology and also commented and improved the manuscript drafts. All authors have read and agreed to the published version of the manuscript.

**Funding:** This research was funded by the European Union through ASIM Scholarship program.

**Data Availability Statement:** The data presented in this study are available on request from the corresponding author. The data are not publicly available due to privacy and ethical reasons.

**Conflicts of Interest:** The authors declare no conflict of interest, and the funders had no role in the design or undertaking of this study.

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
