# Peer review of "The Identification of Sustainability Assessment Indicators for Road Infrastructure Projects in Tanzania"

_sustainability, doi:10.3390/su152014840_

Round 1

Reviewer 1 Report

The work is very relevant, very well written and can be published after minor revisions.

1) L. 123: The authors' names should not appear in parentheses.

2) Table 3: 198.4 billion* (estimated cost).

3) Section 3.3: Insert a space after i), ii), ... vi).

4) L. 221-222: Based on what do the authors conclude that 5 years of experience provides reliable data?

5) In the first row of Table 4 and the first column of Table 5, the sign must be less than or equal (5).

Author Response

Thank you very much for taking the time to review this manuscript. Thank you for the constructive feedback, the authors are appreciative of this. Please find the detailed responses below and the corresponding revisions/corrections highlighted/in track changes in the re-submitted files.

Reviewer 2 Report

The authors recommend specific sustainability indicators suitable for road projects in Tanzania. The paper is well-written and well-understood. Assessment of existing literature well presented. Useful for road projects in Tanzania.

Author Response

(The authors gave the same response as above.)

Reviewer 3 Report

I encourage authors to read carefully the attached file and to consider/correct all the findings and mistakes pointed out there.

English is fine. Minor corrections are required.

Author Response

(The authors gave the same response as above.)
